# The Effect of Silver Nanoparticle Addition on Micropropagation of Apricot Cultivars (*Prunus armeniaca* L.) in Semisolid and Liquid Media

**DOI:** 10.3390/plants12071547

**Published:** 2023-04-03

**Authors:** Cristian Pérez-Caselles, Lorenzo Burgos, Inmaculada Sánchez-Balibrea, Jose A. Egea, Lydia Faize, Marina Martín-Valmaseda, Nina Bogdanchikova, Alexey Pestryakov, Nuria Alburquerque

**Affiliations:** 1Fruit Biotechnology Group, Department of Plant Breeding, CEBAS-CSIC, Campus Universitario de Espinardo, Edif. 25, 30100 Murcia, Spain; cperez@cebas.csic.es (C.P.-C.); burgos@cebas.csic.es (L.B.); inmasb98@gmail.com (I.S.-B.); lbremaud@cebas.csic.es (L.F.); marinavalmaseda@gmail.com (M.M.-V.); 2Fruit Breeding Group, Department of Plant Breeding, CEBAS-CSIC, Campus Universitario de Espinardo, Edif. 25, 30100 Murcia, Spain; jaegea@cebas.csic.es; 3Center for Nanoscience and Nanotechnology (CNyN), Campus Ensenada, National Autonomous University of Mexico (UNAM), Mexico City 04510, Mexico; nina@ens.cnyn.unam.mx; 4Research School of Chemistry and Applied Biomedical Sciences, Tomsk Polytechnic University, Tomsk 634050, Russia; pestryakov2005@yandex.ru

**Keywords:** agar culture, AgNPs, clonal propagation, nanobiotechnology, proliferation, temporary immersion system

## Abstract

Silver nanoparticles (AgNPs) are novel compounds used as antimicrobial and antiviral agents. In addition, AgNPs have been used to improve the growth of different plants, as well as the in vitro multiplication of plant material. In this work the effect of AgNPs on in vitro growth of ‘Canino’ and ‘Mirlo Rojo’ cultivars, as well as the leaf ion composition, are studied. Different concentrations of AgNPs (0, 25, 50, 75 and 100 mg L^−1^) were added to two culture systems: semisolid medium with agar (SSM) in jars and liquid medium in temporary immersion system (TIS). Proliferation (number of shoots), shoot length, productivity (number of shoot × average length), leaf surface, fresh and dry weight were measured. Additionally, the silver and other ion accumulation in the leaves were evaluated by inductively coupled plasma optical emission spectroscopy (ICP-OES) analysis. The productivity of ‘Canino’ and ‘Mirlo Rojo’ decreased when increasing the concentration of AgNPs in the semisolid medium. However, the use of AgNPs in the TIS improved the proliferation and productivity of ‘Canino’ and Mirlo Rojo’, increasing biomass production, and the concentration of nutrients in the plants, although these effects are genotype-dependent. TISs are the best system for introducing silver into shoots, the optimum concentration being 50 mg L^−1^ for ‘Canino’ and 75 mg L^−1^ for ‘Mirlo Rojo’. Principal component analysis, considering all the analyzed ions along the treatments, separates samples in two clear groups related to the culture system used. The use of bioreactors with a liquid medium has improved the productivity of ‘Canino’ and ‘Mirlo Rojo’ in the proliferation stage, avoiding hyperhydration and other disorders. The amount of metallic silver that penetrates apricot plant tissues depends on the culture system, cultivar and concentration of AgNPs added to the culture medium. Silver ion accumulation measured in the shoots grown in the TIS was higher than in shoots micropropagated in a semisolid medium, where it is barely detectable. Furthermore, AgNPs had a beneficial effect on plants grown in TIS. However, AgNPs had a detrimental effect when added to a semisolid medium.

## 1. Introduction

Plant virus diseases are dreadful, because losses produced in crop yields and fruit quality can be devastating. However, the main problem is the lack of disease treatment, with the vector control being the main protection strategy [1,2]. The use of metallic nanoparticles (MeNPs) is widely applied in medicine due to their antimicrobial, virucidal, and anticancer activities, among others [3,4,5]. This has generated a great interest in the application of nanoparticles to agriculture, especially as antiviral agents. Nowadays, there is not much information about the interaction between MeNP-virus and the mechanism behind the effect of nanoparticles on plant viruses. MeNPs may inhibit one or more steps in the virus multiplication cycle, therefore avoiding its multiplication and infection of new cells [6,7]. Additionally, MeNPs may induce defense responses in plants, such as higher expression of genes from phytohormone metabolic pathways (salicylic acid, jasmonic acid, abscisic acid, zeatin riboside and brassinosteroids), or trigger the production of oxygen reactive species (ROS) and antioxidants [8].

Among MeNPs the most used, due to their viricide activity, are silver nanoparticles (AgNPs). AgNPs have been sprayed as a preventive treatment, reducing the accumulation of viral particles and symptoms of *Bean yellow mosaic virus* (BYMV) in the faba bean [9], and *Tomato mosaic virus* in tomato [10]. In different studies, AgNPs have been sprayed on leaves after the inoculation of the virus and, in these cases, they have reduced or completely eliminated symptoms of *Tomato spotted wilt virus* (TSWV) in potato [11], *Banana bunchy top virus* (BBTV) in banana [12] or *Sunhemp rosette virus* (SHRV) in bean [13]. However, as far as we know, AgNPs have not been used to produce virus-free plants.

In this work, the AgNP formulation used was Argovit^®^-7. Viricide activity of Argovit^®^ has been demonstrated against *Canine distemper virus* (CDV) in dogs [14], *White spot syndrome virus* (WSSV) in shrimp [15], *Infectious bronchitis coronavirus* (IBV) in chicken [16], *Rift valley fever virus* (RVFV) in cell cultures and in mouse models [17], and SARS-CoV-2 in preventive treatments in healthy humans [18].

In plants, the interaction between AgNPs and pathogens is only possible if nanoparticles penetrate the plant tissues. Pérez-Caselles et al. [19] demonstrated a large increase in metallic silver concentration within shoot tissues of ‘Canino’ and ‘Mirlo Rojo’ apricot cultivars when Argovit^®^-7 was added to a liquid culture medium, without chloride in its composition, and shoots were cultured in temporary immersion systems (TISs).

In general, plant in vitro culture in the TIS consists of the periodical immersion of micropropagated shoots in a liquid medium, followed by draining of the medium. In order to do that, air pressure is introduced through 0.22 µm air filters, both to impulse the medium upwards and later to add gravity to send the medium back and ensure a uniform immersion time. This type of bioreactor has been used to improve micropropagation and biomass production in different plant species, avoiding hyperhydration or asphyxia common in liquid cultures [20].

The objectives of this work have been: (1) to compare the micropropagation of two apricot cultivars in liquid and in semisolid media, (2) to optimize Argovit^®^ concentration in the culture medium to introduce the maximum possible silver within apricot shoots, (3) to study silver and other ion content in apricot shoot leaves, as affected by Argovit^®^-7 concentration and the different culture systems, and (4) to study the effect of Argovit^®^-7 on apricot micropropagation. 

## 2. Results

### 2.1. Effect of AgNP Concentration on Silver Content within the Plant

The amount of silver that an apricot plant can incorporate into tissues depends on the cultivation system and AgNP concentration in the culture media (Figure 1). Statistical analyses determined significant differences (*p* < 0.001) between cultivars and cultivation systems, and a prediction equation was calculated for the silver content within the plant for each variable (Appendix A). Regarding semisolid media, ‘Canino’ presents a linear adjustment with the highest content of 5.0 mg kg^−1^ of metallic silver at 75 mg L^−1^ AgNPs, while ‘Mirlo Rojo’ shows a cubic adjustment with the highest content at 4.6 mg kg^−1^ of metallic silver at 100 mg L^−1^. However, metallic silver incorporated was between 6.7- and 11.3-fold higher for ‘Canino’, and 5.4- and 9.0-fold higher for ‘Mirlo Rojo’ in the TIS than in semisolid media treatments. Metallic silver introduction in ‘Canino’ and ‘Mirlo Rojo’ cultivars perform similarly in TIS, showing a quadratic adjustment with a maximum between 50 and 75 mg L^−1^ AgNPs. ‘Canino’ was able to incorporate 35.7 mg kg^−1^ of metallic silver at 50 mg L^−1^ AgNPs, and ‘Mirlo Rojo’ 35.5 mg kg^−1^ of metallic silver at the AgNPs concentration of 75 mg L^−1^.

### 2.2. Effect of AgNP Concentration on Plant Mineral Composition

The effect of the addition of different AgNP concentrations to semisolid and liquid media on macro- and micronutrient absorption in ‘Canino’ and ‘Mirlo Rojo’ has been studied.

The concentration of most ions in ‘Canino’ and ‘Mirlo Rojo’ is higher in TISs than in the semisolid medium, with the exception of B and Ni (Appendix A). Plants cultured in TISs with the addition of AgNP at different concentrations showed a similar pattern for some of the ions: P, Ca, Mg, S, Cu, Fe, Mo and Zn for ‘Canino’, and P, S, Cu, Mo, Mn and Zn for ‘Mirlo Rojo’. There was an increase in the absorption of those ions with the increase in the AgNP concentration up to a maximum at 75 mg L^−1^ in the case of ‘Canino’, and at 50 mg L^−1^ (75 mg L^−1^ in the case of Mn) in the case of ‘Mirlo Rojo’, and a decrease afterwards. This trend is very similar to the absorption of Ag ions. On the other hand, the B ion follows the opposite trend, decreasing with a minimum at 75 mg L^−1^ for ‘Canino’ and at 50 mg L^−1^ for ‘Mirlo Rojo’, and then increasing. In a semisolid medium, variability in ion concentration is much smaller than in TIS, although in ‘Mirlo Rojo’ all ion absorption, especially K, P and Mg, decreased (Appendix A), whereas in ‘Canino’, the same happened for most of them (K, Ca, Mg, S, Cu, Fe and Mo) and follow a slightly negative trend with the increase in AgNP concentration (Appendix A). 

A principal component analysis based on mineral contents was carried out in both cultivars. The first two components explain 76.9% of the variability in ‘Canino’ (58.2% PC1 and 18.7% PC2). Figure 2A shows that the first component is related to high concentrations of Mo, Mg, Zn, S, Fe, Ca, Ag, Mn, or even Cu ions, and low concentrations of Ni and B. The second component is related to high concentrations of P and K. Figure 2B groups measurements within the culture system, and there is a very clear separation between TIS and semisolid media. In TIS, they tend to be positive for PC1, especially for the 75 mg L^−1^ treatment, therefore there are high concentrations of ions for this component. On the other hand, measurements in semisolid medium are negative for PC1, and are therefore related to high concentrations of B and Ni. These measurements are distributed along the vertical axis and those from control treatments are the most related to high P and K concentrations.

In ‘Mirlo Rojo’, the two first components explain 69.3% of the variability (53.8% PC1 y 15.5% PC2). Figure 3A shows that PC1 is related to high concentrations of Zn, Fe, Ca, Mg, Mn, Ag, S, Mo, and even Cu, and low concentrations of B and Ni. PC2 is related to high concentrations of P and low K, although the contribution of K is low. Figure 3B shows a clear separation between measurements in TISs and semisolid media. In TIS, there are high concentrations of ions in PC1, as mentioned above, whereas in semisolid, there are low concentrations of ions related to PC1 and higher concentrations of B and Ni.

There are some similarities between PCs in both cultivars. A similar percentage of variability is explained by the first two components. Additionally, PC1 and -2 have high concentrations of the same ions in both cultivars. The main difference is in the K concentration, since in ‘Canino’, it is related to high concentrations in PC2, whereas in ‘Mirlo Rojo’, it is the opposite. Moreover, measurements in each culture system have been grouped on the right in the case of TIS, and on the left in the case of semisolid medium for PC1, this being the most evident difference in ‘Mirlo Rojo’.

### 2.3. Effect of AgNP Concentration on Apricot Micropropagation

Significant differences were observed for most of the variables measured and for their interactions (Appendix A). Consequently, the effect of AgNPs on the micropropagation of each cultivar in each cultivation system was studied separately.

Micropropagation of ‘Canino’ (Figure 4) and ‘Mirlo Rojo’ (Figure 5) was evaluated both in a semisolid medium and in TIS, for AgNP concentrations at 0 to 100 mg L^−1^. A trend analysis of the effect of AgNP concentration on the micropropagation variables was carried out and a prediction equation (Appendix A) was calculated for those variables in which significant differences between treatments were found in media LM−CaCl_2_ and SSM−CaCl_2_ (Table 1). Additionally, micropropagation variables measured in our standard micropropagation media, without AgNPs, are depicted in bars in both Figure 4 and Figure 5, and a contrast analysis was performed to determine if there were significant differences between the cultivation systems.

Micropropagation of both cultivars was better in LM than in SSM (Figure 4 and Figure 5, bars). Significant differences were found for the micropropagation variable productivity (*p* < 0.001) in both cultivars due to an increase in shoot length (being especially relevant in ‘Canino’) and the number of new shoots. Productivity increased 2.3- and 2.0-fold for ‘Canino’ and ‘Mirlo Rojo’, respectively. Fresh, dry weight and leaf area also showed significant differences (*p* < 0.001) for both cultivars. There was a larger increase in the shoot biomass in TISs than in semisolid media, considering that the fresh weight, dry weight and leaf surface were 3.3-, 2.1- and 2.0-fold, respectively, in ‘Canino’, whereas they were 2.4-, 1.9- and 3.2-fold, respectively, in ‘Mirlo Rojo’.

AgNP concentration increase in culture media (LM−CaCl_2_) produced an improvement in the micropropagation in TIS, whereas it was reduced in both cultivars in semisolid media (Figure 4 and Figure 5, lines). Productivity in TISs increased following a linear trend with the increase in AgNP concentration. The best result was obtained at 100 mg L^−1^, being 2.2- and 1.8-fold larger than the control without AgNPs for ‘Canino’ and ‘Mirlo Rojo’, respectively. This increase was mainly due to the same positive trend in proliferation. In both cultivars, the increase in AgNP concentration produced a biomass improvement, as shown by the positive linear trend, followed by fresh and dry weight. 

Productivity in semisolid media (SSM−CaCl_2_) decreased as the AgNP concentration was increased, following a linear trend, due to the same negative linear trend in proliferation in both cultivars. The AgNPs prevented the growth and biomass production of shoots, as shown by the negative linear trend, followed by dry weight in both cultivars. The effect of AgNP concentration on the appearance of ‘Canino’-micropropagated shoots in both systems can be observed in Figure 6A,B.

The TIS produced a micropropagation increase in both apricot cultivars, although this effect was more pronounced for ‘Canino’. The main dissimilarity between cultivars resided in the shoot length average. ‘Mirlo Rojo’ new buds did not elongate as well as ‘Canino’ in order to be used as new shoots. This made productivity much better in ‘Canino’ than in ‘Mirlo Rojo’. The TIS also produced shoots larger and with more biomass, as compared with semisolid media. The dry matter was approximately doubled in both cultivars, although the ‘Canino’ fresh weight was much higher (237% vs. 144% increase). However, ‘Mirlo Rojo’ increased its leaf surface more than ‘Canino’ (217% vs. 103%). As expected, culturing apricot shoots in TISs induced a higher water content in tissues than the semisolid medium, as reflected by a higher relation of fresh/dry weight increase.

The LM−CaCl_2_ and SSM−CaCl_2_ were the culture media used to study the effect of AgNPs in apricot micropropagation. These culture media were modified from the optimized apricot culture media (LM and SSM). Therefore, micropropagation of both cultivars was affected by this change, since the controls without AgNP addition had lower values than the standard apricot culture media, LM and SSM. This effect was more pronounced in the case of liquid media, causing a 44.2% loss of productivity in ‘Canino’ and 67.2% in ‘Mirlo Rojo’.

Regarding the AgNP addition to the culture media, its effect was extremely distinct in the two cultivation systems examined, but with a similar trend for both cultivars. An increase in productivity and biomass production was found when adding the AgNPs to the liquid media, due to the great proliferation of new buds, whereas the AgNPs produced the inhibition of micropropagation in semisolid media.

### 2.4. Rooting and Acclimatization

The effect of adding AgNPs to the culture media was evaluated during the rooting and acclimatization phases. Only shoots coming from TISs were studied due to the negative effect that AgNPs produce on apricot micropropagation when added to semisolid media. Standard rooting protocols in semisolid media were used with the shoots exposed to AgNPs during the proliferation stage.

After 4 weeks of culture in LM−CaCl_2_, ‘Canino’ and ‘Mirlo Rojo’ shoots of all AgNP concentration treatments were transferred to jars with the rooting media. The rooting percentage was recorded after 4 weeks (Figure 6C). A high percentage of rooting shoots was observed for shoots coming from all AgNP treatments (between 87.5 and 100%) in both cultivars. These rooting percentages were not different from those obtained using our standard conditions in LM (90.0%).

Rooted ‘Canino’ and ‘Mirlo Rojo’ shoots from all treatments were acclimatized using standard procedures in a greenhouse, with controlled temperature and light conditions (Figure 6D,E). No differences in survival rates were found independently of the AgNP treatment (percentages between 64.3 and 78.6%) when compared to the control plants (70.0%).

## 3. Discussion

### 3.1. Using Temporal Immersion Systems for Apricot Micropropagation

Semisolid media are the most commonly used culture system in micropropagation. Efficient protocols have been described for the in vitro culture of apricot [21,22]. However, it is always interesting to improve plant material quality and multiplication rates by exploring other micropropagation methods.

Temporary immersion systems (TISs) are efficient culture methods for large-scale micropropagation in liquid media. TISs are considered more effective than the traditional culture in semisolid media, producing better multiplication ratios and biomass. This is due to the explant being completely exposed to the medium, which involves better medium component availability and, therefore, more nutrient absorption [23]. Additionally, air renovation inside the bioreactors helps improve the plants’ physiological state and promote photoautotroph behavior [20,24].

The most popular bioreactor model in woody plants’ micropropagation is RITA^®^ [20,23]. This system is composed of a container divided into two compartments, where plants and liquid media are placed separately. A source of filtered air sends the liquid upwards into the compartment where the plants are, and then, when the air stops, gravity returns the medium to its original place. Many authors have optimized protocols using RITA^®^ for different species, and have compared the proliferation with that obtained in a semisolid medium, reporting an increase in shoot length and formation of new shoots in eucalyptus (*Eucalyptus globulus*) [25], teak (*Tectona grandis*) [26], pistachio (*Pistacea vera*) [27], chestnut (*Castaneae sativa*) [28] and alder (*Alnus glutinosa*) [29], among others. All these authors agree that the use of RITA^®^ produces a high percentage of hyperhydrated plants due to the contact of explants with a liquid medium [30]. To solve this problem, Vidal et al. [28] used stone wool trays to keep chestnut explants in a vertical orientation. Quiala et al. [26] described that a reduction in cytokinin concentration in the medium reduced hyperhydration in teak. However, the most effective way to control this problem is to reduce the time of explant immersion in the liquid medium by reducing the time and/or frequency of immersion [29].

The commercial bioreactor, Plantform^TM^, is also frequently used for TIS. These bioreactors have the same functional concept as RITA^®^ bioreactors, but they have an additional way of introducing filtered air directly to the compartment where the plants are placed, thus reducing the humidity of the explants [31]. Dry air supply to the bioreactor reduces hyperhydration in TISs [28,29,32]. Plantform^TM^ bioreactors have been used to micropropagate myrtle (*Myrtus communis*) [33], goji (*Lycium barbarum*) [32] and cannabis (*Cannabis sativa*) [34], among other species. Some authors have compared proliferation in both systems, RITA^®^ and Plantform^TM^, reporting better multiplication rates and an increase in shoot length in willow (*Salix viminalis*) [35] and alder [29] using Plantform^TM^.

Since apricot is a very sensitive species to hyperhydration [36], the Plantform^TM^ bioreactor was chosen to optimize the micropropagation protocol using TIS. Culture conditions have to be optimized for each species and cultivar to avoid hyperhydration [37,38]. Pérez-Caselles et al. [19] established the conditions to proliferate the apricot cultivars ‘Canino’ and ‘Mirlo Rojo’, which produced healthy explants without hyperhydration, consisting of 2 min of immersion every 6 h, 3 min of aeration every 3 h, 500 mL of culture medium and initial density of 40 explants.

In this work, propagation of ‘Canino’ and ‘Mirlo Rojo’ in TISs and the traditional semisolid medium have been compared observing an improvement in the proliferation of both cultivars, increasing both shoot length and proliferation rates in TIS. Additionally, TISs have increased biomass and shoot vigor. Other apricot cultivars, ‘Ordubad’, ‘Shams’ and ‘Qaysi’, have been propagated before by the TIS, but using non-commercial bioreactors [39]. These authors reported that the TIS improved the micropropagation of those cultivars as compared to proliferation in semisolid medium, by increasing shoot length, leaf number and new shoots. RITA^®^ bioreactors have been used to improve the micropropagation of ‘Myrobolan’ (*Prunus cerasifera*), reporting a faster and more efficient propagation than in a semisolid medium [40]. These results suggest that the Prunus species, including apricot, should be efficiently propagated by TISs.

Plantform^TM^ bioreactors are difficult to use in the rooting phase, since roots can enter holes of the supporting tray, which may involve roots breaking and more difficult management [27,29,41]. To avoid these expected problems, shoots proliferated in TISs were rooted in semisolid medium. Rooting and acclimatization rates were similar to those of shoots proliferated in semisolid media.

### 3.2. Effect of AgNP Concentration on the Penetration of Metallic Silver and Other Ions into Plant Tissue

The use of AgNPs is increasing due to their ability to eliminate phytopathogens which infect plants, such as viruses [42]. The spray of AgNPs at different concentrations on several plant species has been carried out to control plant viruses, reducing virus replication or avoiding the presence of symptoms [9,10,11,12,13]. An interesting application of AgNPs is to obtain virus-free plants. To achieve this purpose, AgNPs added to in vitro culture media should be taken up by plants.

Pérez-Caselles et al. [19] demonstrated that a significant increase in the concentration of metallic silver was achieved in ‘Canino’ and ‘Mirlo Rojo’ shoots by adding 100 mg L^−1^ Argovit^®^ in liquid medium without chlorides in its composition. The LM3 medium (named in this work as LM−CaCl_2_) induced the highest silver incorporation. In this work, the amount of metallic silver incorporated in ‘Canino’ and ‘Mirlo Rojo’ tissues was analyzed as a function of the concentration of AgNPs in LM−CaCl_2_ media, with the aim of maximizing the amount of metallic silver incorporated into the plant.

Metallic silver concentration performed similarly in both cultivars, showing a quadratic tendency with a maximum at the AgNP concentration of 50 mg L^−1^ for ‘Canino’ and 75 mg L^−1^ for ‘Mirlo Rojo’, decreasing silver content at higher AgNP concentrations. In agreement with our results, Castro-González et al. [43] demonstrated silver penetration into stevia (*Stevia rebaudiana*) tissue by using a TIS. They reported that metallic silver content retained in shoots depended on the AgNP (Argovit^®^) concentration, with 200 mg L^−1^ being the most successful treatment. 

The amount of metallic silver incorporated into ‘Canino’ and ‘Mirlo Rojo’ tissues as a function of AgNP concentration in SSM−CaCl_2_ was also studied in this work. Although the SSM−CaCl_2_ is a semisolid medium with agar, it does not contain chlorides in its composition. Therefore, it allows for comparison with the effect of the liquid culture on the penetration of metallic silver into the plant tissue. Significant differences were observed in both cultivars, reaching an increase in the metallic silver concentration at the higher treatment concentration with respect to the control. However, this increase is derisory when compared to that achieved with the liquid medium.

The failure of the semisolid medium may be due to different reasons. On one hand, the movement of AgNPs can be affected by the porous agar matrix size. The hydrodynamic diameter of AgNPs is 56.1 nm, while a concentration of 7 g L^−1^ agar can result in pore sizes of about 100 nm [44]. Moreover, the results of a TEM study showed that AgNPs agglomerate into groups with a size of more than 100 nm, preventing their diffusion through the agar matrix. Additionally, TISs allow for the irrigation of the entire shoot, so that there is a larger shoot contact surface with the culture media. Furthermore, there is evidence that silver AgNPs are able to penetrate into the plant through the leaves when they are applied as a spray [10,45].

‘Canino’ and ‘Mirlo Rojo’ PCAs (Figure 2 and Figure 3) show that plants cultured with AgNPs in TISs tend to exhibit higher concentrations of most ions than when cultured in semisolid media, with the exception of boron and nickel. Plant ion accumulation is improved with the use of a liquid medium, which facilitates a good absorption by all plant tissues [23]. The concentration of the ions increased with the concentration of silver, but this effect was genotype-dependent. Other authors have also reported that AgNP treatment tends to improve macro- and micronutrient accumulation. Spinoso-Castillo et al. [46] informed that AgNPs increased nitrogen and magnesium concentrations and decreased boron concentration in vanilla (*Vanilla planifolia*). AgNP treatments also increased nitrogen and magnesium concentrations in stevia [43], and nitrogen, magnesium and iron concentrations in sugarcane (*Saccharum* spp.) [47]. However, AgNP treatments in semisolid media are inversely correlated with most ion concentrations, except with boron and nickel, which are directly correlated. Boron and nickel have an extremely narrow range between deficiency and toxicity. An inadequate boron supply exhibits a detrimental effect on plants, such as chlorosis and necrosis spreading from the leaf tips, delayed foliation, or a reduction in stem length and dry weight [48]. Nickel’s toxic effects on plants include alterations in stem and leaf growth, as well as biomass production [49].

### 3.3. Effect of AgNP Concentration on Apricot Micropropagation 

Biostimulants are compounds that induce a physiological response in the plant, improving growth and development [50]. Many of them induce a beneficial effect when they are exposed to a low dose, but cause phytotoxicity at a high dose. This process is called hormesis or the hormetic effect [51]. The hormetic effect is often shown in plants exposed to low concentrations of toxic metal ions [52], and some authors have reported the hormetic effect that AgNPs have on several species by using TISs as a cultivation system. Castro-González et al. [43] described an improvement in the micropropagation and stem length of stevia at 50 mg L^−1^ of Argovit^®^, observing phytotoxicity in the plant at 200 mg L^−1^. Spinoso-Castillo et al. [46] informed about a better proliferation and shoot length in vanilla between 25 and 50 mg L^−1^ of Argovit^®^, inhibiting this effect from 100 mg L^−1^. Regarding sugarcane, proliferation stimulation and stem length were observed at 50 mg L^−1^ of Argovit^®^, becoming phytotoxic at 200 mg L^−1^ [47].

Our results proved that AgNPs also produced a hormetic effect on apricot plants. Although AgNPs had a more considerable effect on ‘Canino’ than ‘Mirlo Rojo’, micropropagation rates of both cultivars benefited from the addition of AgNPs in the liquid medium. Improved micropropagation rates in TISs can be explained by an increased accumulation of nutrients in the apricot plants treated with AgNPs. The best productivity of both cultivars was obtained at 100 mg L^−1^. AgNPs considerably increased the proliferation of ‘Canino’ and ‘Mirlo Rojo’, due to the production of new shoots. Nevertheless, differences in stem length were not found. Besides improving micropropagation, AgNPs increased the biomass of ‘Canino’ and ‘Mirlo Rojo’, observing larger fresh and dry weight, with 100 mg L^−1^ being the concentration where the highest values were reached for both cultivars. This effect was also found at 50 mg L^−1^ in vanilla [46] and at 100 mg L^−1^ in stevia [43].

It has been claimed that low concentrations of toxic metals induce hormetic effects through the activation of plant stress defense mechanisms [53]. The hormetic effect and enhanced micropropagation observed in apricot plants may be due to the production of reactive oxygen species (ROS) and the consequent production of phenolic compounds and the activation of antioxidant systems [54]. High concentrations of AgNPs inhibited the antioxidant capacity in vanilla at 100 mg L^−1^ [46] and sugarcane at 200 mg L^−1^ [47], and in these treatments, the plants showed phytotoxicity symptoms. In our case, the best micropropagation ratios were obtained at 100 mg L^−1^, so the point of inhibition of antioxidant capacity has not yet been reached. 

The effect of AgNPs on ‘Canino’ and ‘Mirlo Rojo’ micropropagation in the semisolid medium was also analyzed. AgNPs had a toxic effect on this cultivation system, since both micropropagation and biomass production were inhibited in the two cultivars. The ratios measured decreased, while AgNP concentration increased in the medium. This effect was accentuated from 25 mg L^−1^. Additionally, the high accumulation of boron and nickel in a semisolid medium could be related to detrimental effects on micropropagation. Our results contrast with those obtained by Pastelín-Solano et al. [55]. They reported that Argovit^®^ added to a semisolid medium caused a hormetic effect in vanilla, improving proliferation, stem length and biomass production at 50 mg L^−1^, and causing phytotoxicity from 100 mg L^−1^. However, they used Phytagel^TM^ as a gelling agent in the medium, meanwhile ours was agar. The type and concentration of gelling agent may be essential for the use of AgNPs in a semisolid medium. Phytagel^TM^ has macropores with the size between 100 and 500 µm [56], and has been described to encourage faster plant growth, as compared with agar [57].

Apricot shoot rooting was carried out in a semisolid medium due to the problems that TISs lead to in this phase, which were explained in the prior section. This paper has only studied the effect of AgNPs on the proliferation phase. Therefore, the rooting of the shoots has been made by the conventional protocol in order to check that AgNPs did not harm rooting. Shoots multiplied in the presence of every AgNP concentration were rooted in the usual way, finding no significant differences with respect to the control. Likewise, rooted shoots from all the treatments were acclimatized following standard procedures.

## 4. Materials and Methods

### 4.1. Plant Material

Micropropagated ‘Canino’ shoots infected with *Plum pox virus* and ‘Mirlo Rojo’ shoots infected with *Hop stunt viroid* are used in this study, since these constitute excellent model apricot plants to study the possible effect of AgNPs on the pathogen, and also on micropropagation. Both cultivars were maintained in 500 mL glass jars, in a culture chamber at 23 ± 1 °C, with a photoperiod of 16/8 h light/darkness (56 μmol m^−2^ s^−1^).

### 4.2. Plant Multiplication in Semisolid and Liquid Medium

‘Canino’ and ‘Mirlo Rojo’ shoots were multiplied in semisolid medium (SSM) or liquid medium (LM), as previously described [19,22]. Briefly, the SSM consisted of QL macronutrients [58], DKW micronutrients and vitamins [59], 3 mM calcium chloride, 3% (*w*/*v*) sucrose, 0.7% (*w*/*v*) agar, 0.8 mM phloroglucinol, 1.12 µM 6-benzylaminopurine riboside (BAPr), 0.05 µM indole-3-butyric acid (IBA), 2.1 µM 6-(3-hydroxybenzyl amino) purine (meta-topolin) and 29.6 µM adenine. LM had the same composition as SSM without agar addition. The medium pH was adjusted to 5.7, before autoclaving at 121 °C for 20 min. Micropropagated shoots were transferred to a fresh medium every 4 weeks.

Apricot micropropagation with LM was performed in TISs using 4 L Plantform^TM^ bioreactors. Of liquid medium, 500 mL was poured into each bioreactor after autoclaving. The pressure air was introduced into the bioreactor through 0.22 µm air filters and the growing conditions were 2 min of immersion every 6 h, and 3 min of aeration every 3 h. The medium was refreshed and plant material was subcultured every 4 weeks.

### 4.3. Addition of AgNPs to the Medium

AgNPs used in this work, formulated as Argovit^®^-7, were kindly donated by the Scientific Production Centre Vector-Vita Ltd., Novosibirsk, Russia. The physicochemical characteristics of this formulation have been described previously by Pérez-Caselles et al. [19]. Briefly, Argovit^®^-7 is an aqueous solution with 1.2% (*w*/*w*) of metallic silver and 18.8% (*w*/*w*) of polyvinylpyrrolidone (PVP), with an AgNP concentration of 200 g L^−1^.

Different concentrations of AgNPs (0, 25, 50, 75, and 100 mg L^−1^) were added to modified SSM or LM, after being autoclaved (Table 1). The modification consisted of the elimination of calcium chloride (SSM−CaCl_2_ or LM−CaCl_2_) to allow for the incorporation of silver into plant tissues, as described by Pérez-Caselles et al. [19].

### 4.4. Rooting and Acclimatization of Plants Grown in a Liquid Medium with AgNPs

The effect of adding AgNPs to the liquid media during micropropagation was evaluated in the rooting phase, which was carried out in semisolid media. After 4 weeks of culturing in LM−CaCl_2_ with AgNPs (25, 50, 75, or 100 mg L^−1^), 16 apricot shoots of at least 1.5 cm length from each treatment were cultured in two jars with rooting media. Previously, the base of the shoots was immersed in 4.9 mM IBA to induce rooting and the shoot apexes were soaked with 44.4 µM BAPr to avoid necrosis. ‘Canino’ and ‘Mirlo Rojo’ shoots were rooted in RM1 and RM2, respectively (Table 1). RM1 was described by Pérez-Tornero and Burgos [21], with modifications. Briefly, it consisted of QL macronutrients, DKW micronutrients and vitamins, 1.5 mM calcium chloride, 2% (*w*/*v*) sucrose, 0.7% (*w*/*v*) agar, 0.8 mM phloroglucinol, 3 µM IBA and 29.6 µM adenine. RM2 was based on RM1 by adding 17.1 µM indole-3-acetic acid (IAA) and 87 mg L^−1^ ethylenediamine di-2-hydroxyphenyl acetate ferric (Fe-EDDHA). The medium pH was adjusted to 5.7, before autoclaving at 121 °C for 20 min. After 4 weeks, the percentage of rooting was evaluated.

Rooted plants were acclimatized following standard procedures for apricots [21]. Roots were washed with water to remove the remaining agar and plants were transferred to 300 mL pots, containing a mixture of peat and perlite (2:1 *v*/*v*), and fertilized with 20 mL of the Hoagland solution [60]. Plants were placed in the greenhouse under controlled and natural daylight conditions. The survival rates were recorded after 6 weeks.

### 4.5. Micropropagation Evaluation

The effect of the addition of AgNPs to SSM and LM on ‘Canino’ and ‘Mirlo Rojo’ micropropagation was evaluated. Different variables, such as proliferation (number of new shoots longer than 1 cm from each initial shoot), mean length of new shoots longer than 1 cm, productivity (the product of proliferation and new shoot mean length), leaf area measured at the fourth leaf on the main shoot, as well as fresh and dried weight, were recorded in thirty shoots from each apricot cultivar after 4 weeks of culture in SSM−CaCl_2_ or LM−CaCl_2_, supplemented with each AgNP concentration tested. The same variables were measured in ‘Canino’ and ‘Mirlo Rojo’ shoots micropropagated in SSM and LM as the control micropropagation media.

### 4.6. Analysis of Silver Content and Other Minerals in the Plant Tissues

The silver content and other minerals (K, P, Ca, Mg S, B, Cu, Fe, Mn, Mo, Ni and Zn) were obtained using 200 mg of dry material from whole shoots, without basal calli. Three repetitions per sample (‘Canino’ and ‘Mirlo Rojo’ cultured in SSM−CaCl_2_ or LM−CaCl_2_ with AgNPs) were analyzed. Fresh samples were dried out at 60 °C for 72 h. The dried powder was subjected to acid digestion in an HNO_3_/H_2_O_2_ solution (4:1) using an UltraCLAVE microwave. Silver and other mineral concentration was determined by inductively coupled plasma optical emission spectroscopy (ICP-OES) at CEBAS Ionomic Service, using iCAP 7000 (Thermo Fisher Scientific, Waltham, MA, USA).

### 4.7. Statistical Analyses

Results of the micropropagation evaluation were analyzed using SAS version 9.4 (SAS Institute, Cary, NC, USA). An analysis of variance determined if there were significant differences between cultivars, culture systems or AgNP concentrations. Trend analysis and orthogonal contrasts were used to study the effect of AgNP concentration. Results obtained from standard media without AgNP addition for each cultivar were compared.

Principal component analysis, using mineral contents as variables, was applied through the R package *factomineR* [61] for each cultivar, considering both culture media.

## 5. Conclusions

TISs are an alternative to traditional culture in semisolid medium. This study has shown that Plantform^TM^ bioreactors have improved the productivity of ‘Canino’ and ‘Mirlo Rojo’ in the proliferation stage, avoiding hyperhydration and other disorders. Regarding the use of AgNPs, the amount of metallic silver that penetrates apricot plant tissues depends on the culture system, cultivar and concentration of AgNPs added to the culture medium. Silver accumulation measured in the shoots grown in TISs was higher than in the shoots micropropagated in semisolid media, where it is barely detectable. The optimal concentration in a liquid medium to maximize the concentration of silver in the plant was 50 and 75 mg L^−1^ for ‘Canino’ and ‘Mirlo Rojo’, respectively. Furthermore, AgNPs had a beneficial effect on plants grown in TISs, improving proliferation, increasing biomass production and increasing the concentration of nutrients in the plants. However, AgNPs have a detrimental effect when added to a semisolid medium, reducing micropropagation and biomass production.

## Figures and Tables

**Figure 1 plants-12-01547-f001:**
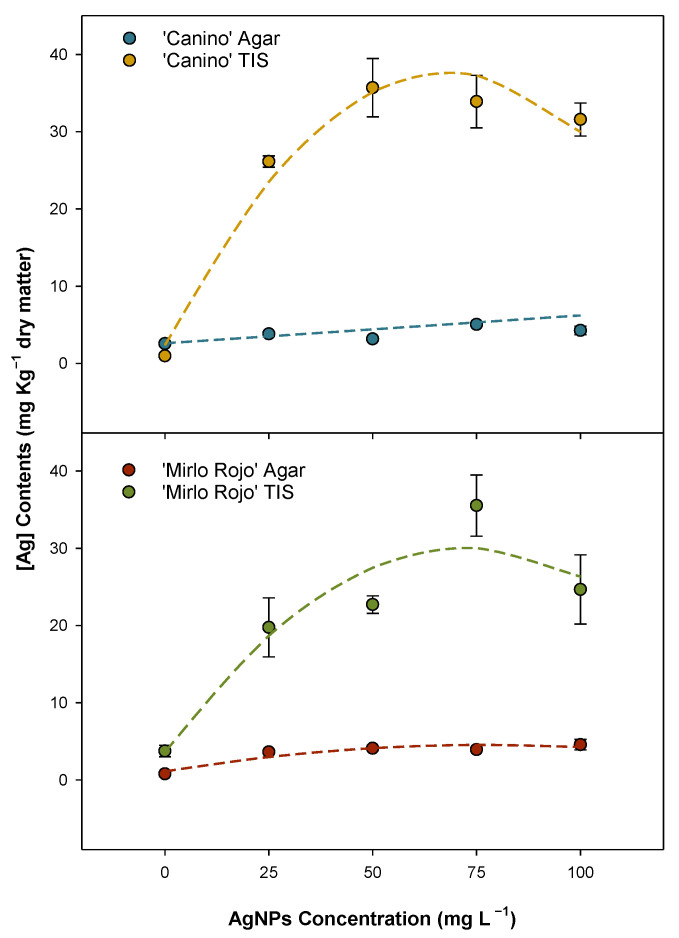
Effect of AgNP concentration increase for ion silver content in ‘Canino’ and ‘Mirlo Rojo’ shoots, cultured in both TISs and semisolid media. Polynomial equations are represented by dashed lines.

**Figure 2 plants-12-01547-f002:**
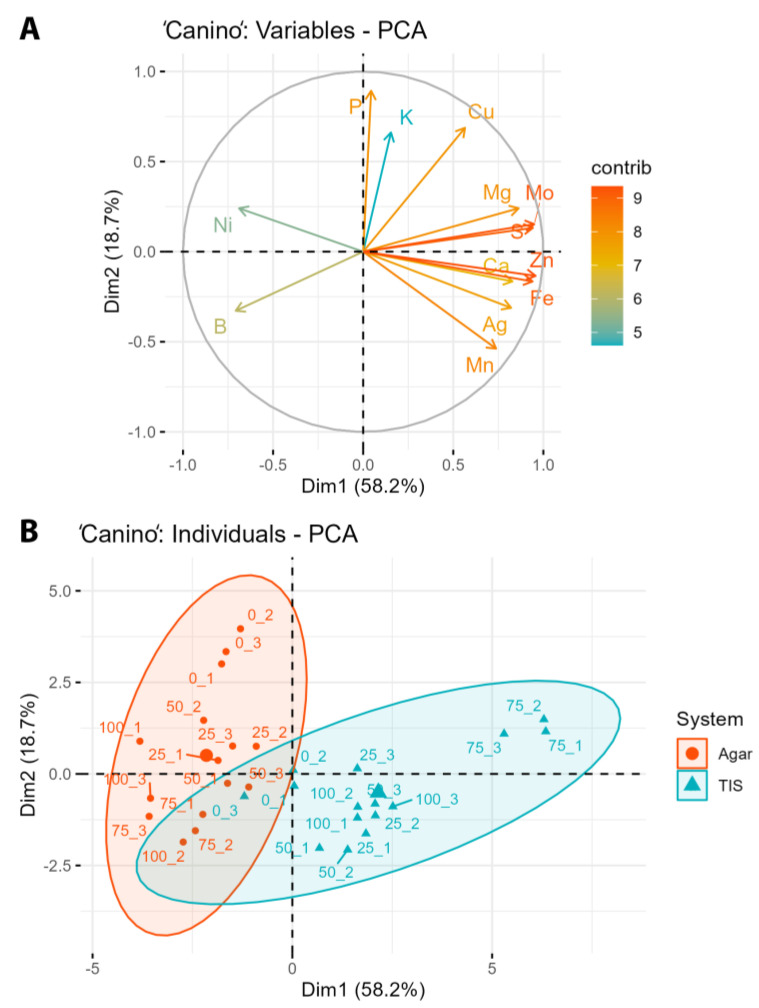
Principal component analysis of nutrient absorption in ‘Canino’, as a function of AgNP concentration and the culture system. (**A**) Analysis of variables, (**B**) analysis of individuals.

**Figure 3 plants-12-01547-f003:**
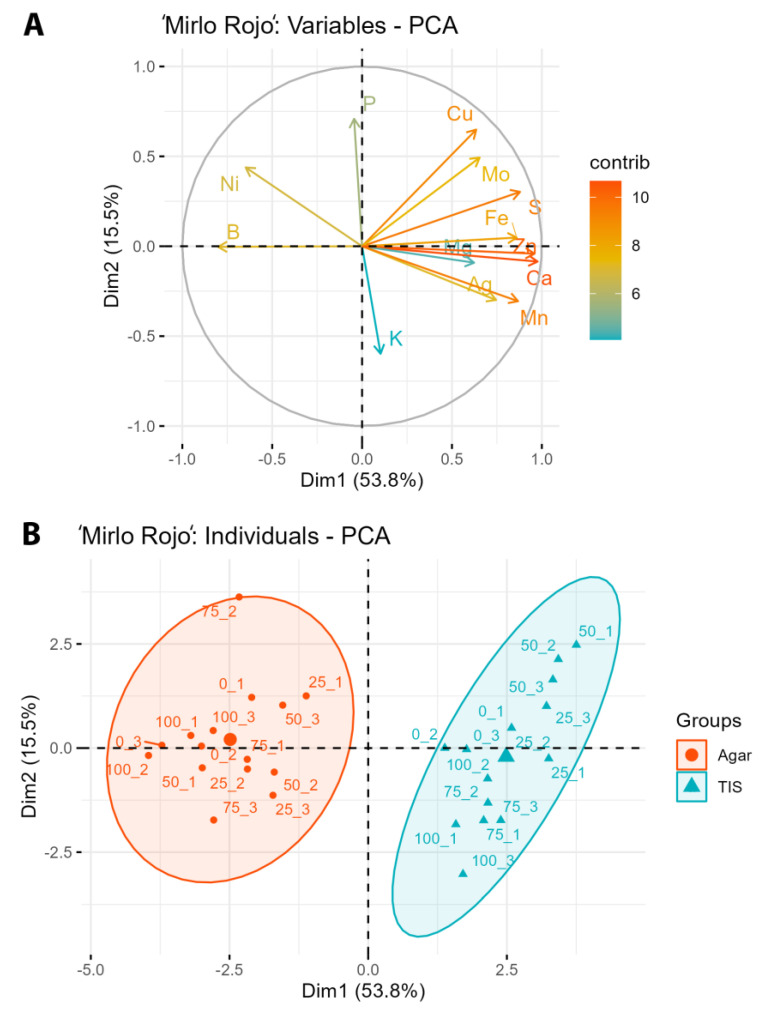
Principal component analysis of nutrient absorption in ‘Mirlo Rojo’, as a function of AgNP concentration and the culture system. (**A**) Analysis of variables, (**B**) analysis of individuals.

**Figure 4 plants-12-01547-f004:**
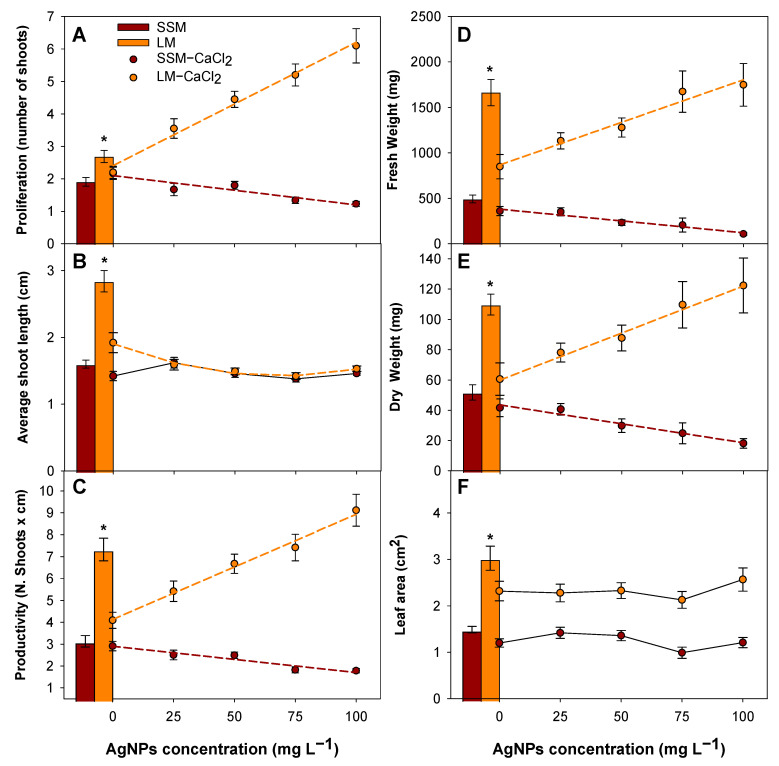
Effect of AgNP concentration on proliferation (**A**), shoot length (**B**), productivity (**C**), fresh weight (**D**), dry weight (**E**) and leaf surface (**F**) of ‘Canino’, both in semisolid media and TIS. When the data followed a significant trend, polynomial equations are represented by dashed lines. Asterisks indicate significant differences between LM and SSM.

**Figure 5 plants-12-01547-f005:**
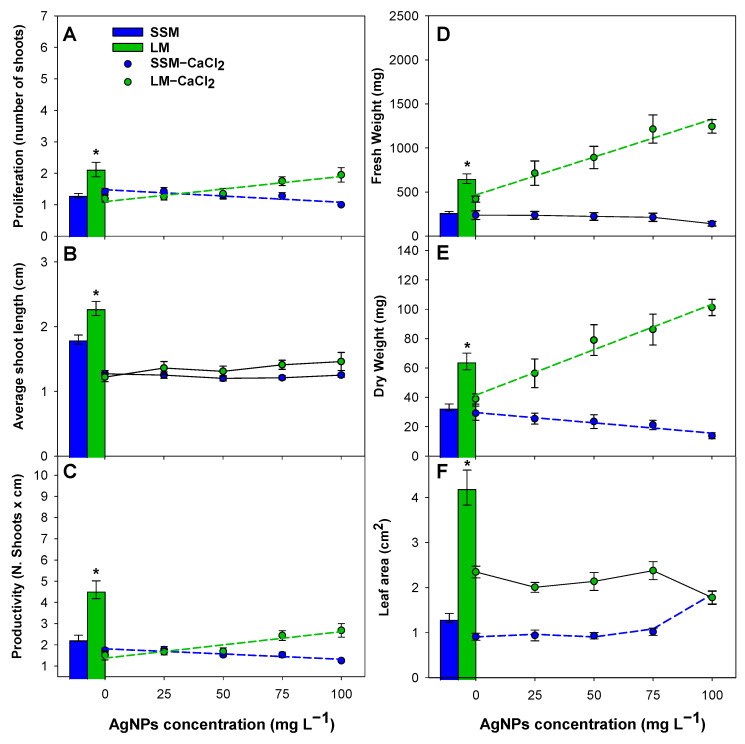
Effect of AgNP concentration on proliferation (**A**), shoot length (**B**), productivity (**C**), fresh weight (**D**), dry weight (**E**) and leaf surface (**F**) of ‘Mirlo Rojo’, both in semisolid media and TIS. When the data followed a significant trend, polynomial equations are represented by dashed lines. Asterisks indicate significant differences between LM and SSM.

**Figure 6 plants-12-01547-f006:**
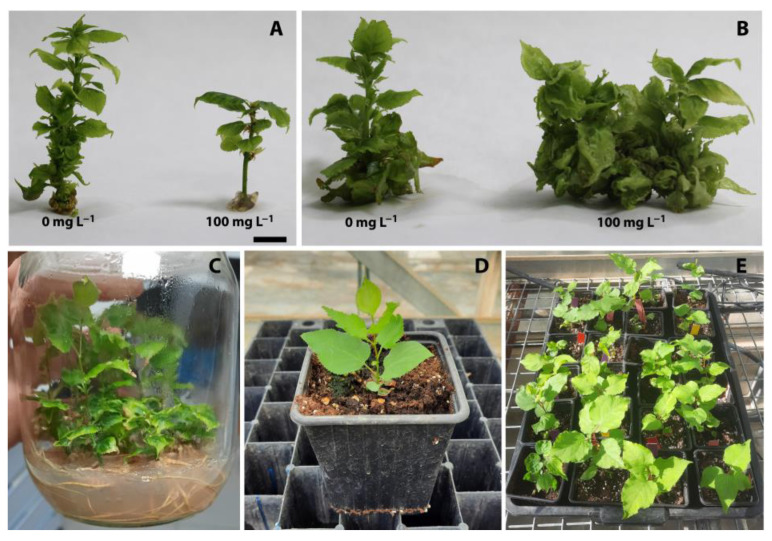
Apricot plants treated with silver nanoparticles (AgNPs). Effect of AgNP concentration on apricot micropropagation of the ‘Canino’ apricot cultivar in a semisolid medium (**A**) and the TIS (**B**). ‘Canino’ plants rooted in vitro (**C**) and acclimatized in the greenhouse (**D**,**E**). The bar represents 5 mm in (**A**,**B**).

**Table 1 plants-12-01547-t001:** Culture media used for apricot micropropagation study.

Media Codes	Basal Salts	CaCl_2_ (3 mM)	Other Components	Agar (7 g L^−1^)
SSM	QL + DKW	Yes	a	Yes
SSM−CaCl_2_	QL + DKW	No	a	Yes
LM	QL + DKW	Yes	a	No
LM−CaCl_2_	QL + DKW	No	a	No
RM1	QL + DKW	½	b	Yes
RM2	QL + DKW	½	c	Yes

a: Phloroglucinol (0.8 mM), 3% sucrose, 1.12 µM BAPr, 0.05 µM IBA, 2.1 µM meta-topolin and 29.6 µM adenine; b: 0.8 mM phloroglucinol, 2% sucrose, 3 µM IBA and 29.6 µM adenine; c: 0.8 mM phloroglucinol, 2% sucrose, 3 µM IBA, 17.1 µM IAA, 87 mg L^−1^ Fe-EDDHA and 29.6 µM adenine.

## Data Availability

Not applicable.

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
