# Peer review of "The Effect of Silver Nanoparticle Addition on Micropropagation of Apricot Cultivars (Prunus armeniaca L.) in Semisolid and Liquid Media"

_plants, 2023, doi:10.3390/plants12071547_

Round 1

Reviewer 1 Report

I recommend the publication of the manuscript " The effect of silver nanoparticles addition on micropropagation of apricot cultivars (Prunus armeniaca L.) in semisolid and liquid medium", written by Cristian Pérez-Caselles, Lorenzo Burgos, Inmaculada Sanchez-Balibrea, Jose A. Egea, Lydia Faize, Marina Martín-Valmaseda, Nina Bogdanchikova, Alexey Pestryakov, and Nuria Alburquerquein. The matter is extremely interesting and innovative for the obtainement of free-pathogen plants by using tissue culture systems.

Very few minor revisions, indicated below, are desirable to improve the readability of the text.

- Lines 64-67. " In our laboratory sharka-infected ‘Canino’ shoots and Hop Stunt Viroid-infected ‘Mirlo Rojo’ shoots are maintained in micropropagation. These constitutes excellent model apricot plants to study the possible effect of AgNPs on the pathogen and also on micropropation." This sentence induces the reader to think that the experimentation will also be conducted on this infected material, which then did not happen. Better to include this concept in the concluding part, as future prospects.

Some typos:    

-Lines 99/100/105/106 the unit for the content of silver ions is not in line with the related figures. In the text the unit is mg, while in the figures is mg Kg -1 of Dry Matter

- Line 273: Eucalyptus globulus instead of Eucalyptus globolos

Author Response

Dear Reviewer,

Thank you so much for all your comments.

Regarding the text in Lines 64-67, we agree that the idea is confusing and is not in the right place. Our experiments were carried out with micropropagated shoots of 'Canino' cv. infected with Plum pox virus and 'Mirlo Rojo' cv. infected with Hop Stunt viroid since these materials are excellent model apricot plants to study the possible effect of nanoparticles on the pathogen and also on micropropagation. We have moved this information to 'Materials and Methods' section (lines 430-433).
We want to apologize for not giving this information about the material with which we have worked clearly. In the corrected version of the article this aspect is clarified.

We have corrected the units for the content of silver ions (mg Kg -1 of Dry Matter  instead mg) in lines 99/100/105/106. 

We have also modified the name in Line 273: Eucalyptus globulus instead of Eucalyptus globolos.

Reviewer 2 Report

The reviewed manuscript described interesting and new data of using the silver nanoparticles addition on micropropagation of apricot cultivars.  The topic of the paper is appropriate for journal Plants. The abstract, key words, and introduction reflect the basic ideas of the manuscript. The authors used the relevant methods, the results obtained are supported by data.  The references list includes the key citations on this topic. It is necessary to note, that the MS is well edited and clear. I am sure that the MS will be interesting to readers.

I can recommend it for publication with the minor revision.

Lines 64-67 and 78-79: These sentences are confusing for me, because here you describe your research methods. Please, modify it or transfer into Materials and Methods section.

 Figures 4, 5. Please, enlarge the figures, because the letters pure visible.

Author Response

Dear Reviewer,

Thank you for all your suggestions.

The lines 64-67 have been removed from Introduction section and placed in 'Material and Methods' section as suggested. Now this information is in lines 430-433.

The lines 78-79 have been eliminated since information is redundant.

Regarding Figures 4 and 5, the letter sizes have been increased.

I hope the new version of the document would be better now.